# The burden of traditional neonatal uvulectomy among admissions to neonatal intensive care units, North Central Ethiopia, 2019: A triangulated crossectional study

**Wubet Alebachew Bayih**[ID]***, Biniam Minuye Birhan, Abebaw Yeshambel Alemu**

Debre Tabor University, Debre Tabor, Ethiopia

* wubetalebachew@gmail.com

## Abstract

### Background

Traditional neonatal uvulectomy is unsupervised, unscientific and potentially dangerous cultural malpractice. It is often accompanied with life threatening neonatal morbidities such as infection, septicemia, anemia, aspiration and oropharyngeal injury. However, there is no current regional and even national data of its public health importance in the health care system. Therefore, this study was aimed at assessing the burden, associated factors and reasons of traditional uvulectomy among neonatal admissions at Debre Tabor General Hospital, North Central Ethiopia, from September 2018 to August 2019.

### Methods

A quantitative cross sectional study supplemented with phenomenological study was employed on 422 mother-neonate pairs. Eight mothers who were not included in the quantitative part were involved as key informants of the qualitative study. Systematic and purposive sampling techniques were used to select study participants for the quantitative and qualitative parts of the study respectively. Multivariable logistic regressions were fitted to investigate significant predictors of traditional neonatal uvulectomy at p-value $\leq$ 0.05 and 95% CI. Moreover, the qualitative data were carefully transcribed, coded, screened, thematized, synthesized and then triangulated with the quantitative results.

### Results

The burden of postuvulectomy admission was 67 (15.88%). Most of these admissions had post uvulectomy sepsis [59 (88.1%)] followed by anemia (55.23%). From multivariable analysis, factors that had significant odds of association with traditional neonatal uvulectomy include: having male neonate [AOR = 4.87; 95% CI: 1.10, 21.59], antenatal couple counseling about traditional neonatal uvulectomy [AOR = 0.053; 95% CI: 0.01, 0.35], home delivery [AOR = 6.02; 95% CI: 1.15, 31.61], postnatal couple counseling about traditional neonatal uvulectomy [AOR = 0.101; 95% CI: 0.02, 0.65], prior history of traditional neonatal

**Data Availability Statement:** All relevant data are within the manuscript and its Supporting Information files.

**Funding:** The author(s) received no specific funding for this work.

**Competing interests:** The authors have declared that no competing interests exist.

**Abbreviations:** CSA, Central Statistical Agency; DTGH, Debre Tabor General Hospital; DTU, Debre Tabor University; EDHS, Ethiopian Demographic Health Survey; NICU, Neonatal Intensive care unit.

uvulectomy [AOR = 7.15; 95% CI: 1.18, 43.21] and knowing at least one adverse effect of traditional neonatal uvulectomy [AOR = 0.068; 95% CI: 0.01, 0.44]. Furthermore, maternal perception of "*there is no modern medicine to treat elongated and swollen neonatal uvula*" was the most explained reason to practice traditional neonatal uvulectomy.

## Conclusion and recommendation

The burden of traditional neonatal uvulectomy was high. Fortunately, its predictors are modifiable. Therefore, several advocacy teams of neonatal health consisting of mainly women health development armies, elders, religious fathers, health professionals and criminal prosecutors should be actively mobilized against traditional neonatal uvulectomy. Besides, parental couple counseling about the adverse effects of traditional neonatal uvulectomy should be properly implemented in the routine antenatal and postnatal continuum of care in South Gondar Zone, North Central Ethiopia.

## Background

Uvula is a small soft tissue that hangs down from the back of the mouth above the throat between the two tonsils [1,2]. It has its own natural advantages of preventing aspiration, lubricating oropharyngeal mucosa, serving for language communication, boosting immunological function and prevention of breast milk regurgitation through the neonatal nose [1, 3, 4].

Traditional neonatal uvulectomy is unscientific, unsupervised and potentially dangerous practice that involves partial or total removal of uvula and sometimes the tonsils using unsterilized traditional instruments (sharp blade, horsetail hair or thread with a loop)[1, 5–7]. These instruments are usually used on several neonates in the same session thereby increasing the transmission of communicable infections mainly HIV and hepatitis. Moreover, other complications of traditional neonatal uvulectomy include anemia, hemorrhage, sepsis, jaundice, septicemia, tetanus, neck infection, pharyngeal dryness, aspiration, pain for many days after the procedure, change in voice, disturbance in sleep pattern, regurgitation of breastmilk from the nostril and cavernous sinus thrombosis[2, 6, 8–10].Neonatal admissions attributed to thesecomplications require antibiotics, oxygen, intravenous fluid, blood transfusion, phototherapy and greater number of health care providers thereby accelerating the cost in the health care system [9, 11–16].

In Africa, neonatal uvulectomy by traditional practitioners has been an age-long practice [1–4, 9, 12, 14, 17–21]. The procedure persists in the developing countries probably because of low socioeconomic status and non-formal educational level [4, 18]. There are divergent views to the reason as well as its overall benefit in these countries [17]. Besides, there have been reported cases of complications after the procedure with a subsequent increase in mortality [6, 7, 11, 22–24]. In the study area, the community mistakenly attributes nearly all neonatal illnesses to uvular swelling and elongation. Thus, ill neonates are often subjected to traditional uvulectomy for misconceived better cure [10, 25]. Educating the community sustainably about the harmful effects of traditional uvulectomy is thought to bring behavioral change in the study area so that it could be possible to reduce neonatal mortality from traditional uvulectomy [12, 26–30].

Ethiopian neonatal mortality rate (30/1000) is among the five highest neonatal mortality rate burden countries in the world. Amhara region, where the study area is located, comprised

the highest proportion of the national burden (47%) [9]. Morbidities from the post uvulectomy complications contributed to the magnificent burden of neonatal mortality in the region. This is because unlike other harmful traditional practices, traditional neonatal uvulectomy is still being practiced in the region mainly at South Gondar Zone, North Central Ethiopia [9, 10]. Nonetheless, there is no current regional and even national data about the burden, associated factors and reasons of this malpractice.

## Methods

### Study setting and period

The study was conducted from September 2018 to August 2019 at Debre Tabor General Hospital, South Gondar Zone, Amhara region, North Central Ethiopia. The hospital is found 666 km far from Addis Ababa and 105 km away from Bahir Dar town. It is the largest hospital in South Gondar zone serving about 2.7 million populations and linked to 7 district hospitals. Neonatal intensive care unit of the hospital had a total of 28 neonatal beds and hosted approximately 987 admissions annually. Other than prematurity and perinatal asphyxia, unknown number of admissions was attributed to the complications of harmful traditional practices like traditional neonatal uvulectomy [10, 25].

### Study design and participant characteristics

Mixed type (quantitative supplemented with qualitative) hospital based cross sectional study was conducted. Phenomenological study design was employed for the qualitative part. There were only 10 admissions excluded of the study (3 abandoned neonates, 5 neonates whose mothers with critical medical illness and 2 neonates whose mothers having confirmed postpartum major depression disorders). The abandoned neonates were excluded because there was no other source of subjective data for these neonates. Moreover, the mothers with critical medical illness and postpartum major depression disorders were not mentally and physically capable of being interviewed.

### Sample size determination and sampling procedure

By using single population proportion formula and considering confidence level /Z/ of 95%, marginal error of 5%, a reasonable estimate for the proportion of traditional neonatal uvulectomy (P = 0.5) and adding a none response rate of 10%, a total sample of 422 mother-baby pairs was obtained. By using systematic sampling, every other eligible mother baby pair admitted to the neonatal intensive care unit was selected for the quantitative part of the study. Sample size for the qualitative part of the study was determined based on saturation of the required information. Saturation was considered when repetitive qualitative responses were generated. Repetitive qualitative responses were reached after interviewing 8purposively selected mothers thereby suggesting suffice of these mothers for the qualitative part. These 8 mothers were not involved in the quantitative study.

### Measurement and data collection procedure

For quantitative part of the study, data were collected by four trained BSc neonatal nurses through face to face interview using a validated and structured questionnaire (S1 File) that was developed from reviewing different studies of traditional uvulectomy and other harmful traditional practices [1–7, 11–21, 26–28]. Interviews were made for eligible mother-neonate dyad sat NICU. The questionnaire contained factors related to maternal socio demography, obstetrics, neonatal health related characteristics and maternal knowledge of uvula and uvulectomy.

Besides, a checklist was employed to abstract data on medical diagnosis and postuvulectomy complications at admission to neonatal intensive care unit.

For qualitative part of the study, in-depth interview was conducted using a semi-structured interview guide (S2 File) during last month of the survey by the principal investigator and one supervisor to supplement the quantitative findings and address issues that were not touched by the quantitative part. At the initial interview, open ended questions were raised about maternal behavioral and cultural perceptions, reasons and experience of traditional neonatal uvulectomy. Then, prompt questions on all aspects of maternal perceptions towards the malpractice were adequately probed by the principal investigator as necessary as listed in the in-depth interview guide. Each in-depth interview was tape recorded and lasted between 30 to 40 minutes.

## Data quality control

The questionnaire was first prepared in English and then translated to local language, Amharic, suitable for data collection. The Amharic version was then retranslated back to English to check for consistency. There was a further possibility that some women did not understand the questions fully, or that difficulties arose in translation. To work on this challenge, two professional translators with medical experience were hired during data cleansing in order to explain the questions and correctly translate maternal answers for minimizing the risk of information loss during translation. The tool was adapted from different studies in Ethiopia [3, 12, 14, 18, 20, 21], Kenya [8], Tanzania [6, 7, 11], Niger [13] and Nigeria [4, 26, 28].

Five days of training (two days theoretical and three days practical) was first provided for data collectors and supervisors about pretesting and the process of data collection. Before the actual data collection, pretest was done on 21 eligible mother-baby pairs (5% of sample size) at Debre Tabor General Hospital2 weeks prior to the study just to evaluate the clarity of questions, validity of the tool and reaction of the respondents to the questions. During data collection, data collectors were closely monitored and guided by two MSc neonatal nurse supervisors for complete and appropriate collection of the data. Reporting of the collected data to the principal investigator was made on a daily basis. Furthermore, the collected data were double entered into Epidata version 4.2 by two data clerks and consistency of the entered data were cross checked by comparing the two separately entered data for validation purpose. Besides, to minimize bias, interviews were conducted in an area with adequate confidentiality and privacy without the involvement of health care providers working in that hospital. Simple frequencies and cross tabulations were done for missing values and crosschecked with hard copies of the collected data.

## Statistical analysis

The double entered data were exported to SPSS version 23 software for data transformation and further analysis. Frequencies, proportion, summary statistics and cross tabulation were used to describe the study population in relation to the study variables and presented in tables. The assumptions for binary logistic regressions were first checked and then bivariable analysis was carried out to identify candidate variables ($p<0.25$) for multivariable analysis. Then, multivariable logistic regression analysis was performed using those candidate variables to investigate statistically significant independent predictors of traditional neonatal uvulectomy. Finally, variables whose p value less than 0.05 ($p<0.05$) from multivariable logistic regression were declared as statistically significant using adjusted odds ratio of 95% CI. Multi-collinearity between the study variables was diagnosed using standard error and correlation matrix. Hoshmer-Lemeshow statistic and Omnibus tests were also performed to test for model fitness. For the qualitative study, the in-depth interviews were transcribed verbatim in Amharic audios

and translated into English by language expert. Data were analyzed using thematic analysis approach. Each transcript was carefully screened and triangulated with the quantitative result.

**Ethical approval and consent to participate.** The authors reached that obtaining only informed voluntary verbal consent was enough for ethical approval by the ethics committee due to the following reasons: I) regarding women's educational status, the authors had prior data indicating that nearly half (48%) of the women in Ethiopia didn't have the ability to read and write [9]. II) The study was an interviewer based crossectional study aimed for the direct beneficence of mothers in improving neonatal health through boosting their awareness towards the adverse health impact of traditional neonatal uvulectomy III) The study didn't also involve any measurement that could bring physical harm to the mothers and their neonates. IV) Each respondent's informed verbal voluntary consent was marked as '√' and recorded in the cover page of hardcopy of the questionnaire and interview guide where it can stay there as long as possible. Therefore, taking all the aforementioned parameters into consideration, the Institutional Health Research Ethics Review Committee (IHRERC) of Debre Tabor University assured ethical approval of the study.

# Results

## Maternal socio-demographic characteristics

All the eligible mothers were participated in the study thereby making 100% response rate. More than three quarters of the respondent mothers were urban residents [324(76.78%)] and nearly similar number of respondents [322(76.30%)] were in the age group of 20–34 years old. Nearly all the mothers [406(96.21%)] were married. More than half of the mothers were primiparous [223(52.84%)]. Regarding educational status, about one fifth of the mothers [76 (18.01%)] and 32 (7.58%) of the husbands were unable to read and write. Moreover, about half of the mothers were civil servants [216 (51.18%)]. More than two third of the mothers [283 (67.09%)] had an average monthly income above poverty line (≥**37.5 $ US**) (**Table 1**).

## Obstetrics related factors

Four hundred five (95.97%) mothers had antenatal care during pregnancy of the index neonate. However, only about two-third of them [277(68.40%)] attended four and above ANC visits. It was about one third of the mothers [125(30.86%)] who were accompanied with their spouses during antenatal care. Regarding counseling of traditional neonatal uvulectomy, one fourth [103 (25.43%)] of the mothers were given antenatal counseling. Besides, 44 (42.72%) of the mothers were given the counseling together with their husbands. Nearly one third [130(32.23%)] of the mothers gave birth at home. Moreover, about 80% of the respondent mothers had at least one post natal care visit and only 46 (13.73%) of whom attended the second postnatal care visit. During postnatal care visit, 317 (94.63%) respondent mothers were accompanied by their spouses. More than one third of the mothers [127 (37.91%)] were counseled of traditional neonatal uvulectomy during their post natal care. However, only 48 (37.80%) of them were given the counseling together with their spouses. The counseling was about adverse effects of traditional neonatal uvulectomy[85 (66.93%)], presence of modern medicine for perceived uvular swelling and elongation[46 (36.22%)], immediate modern health care seeking behavior during maternal perception of uvular swelling and elongation [54 (42.52%)] and the benefits of uvula[22 (17.32%)] (**Table 2**).

## Neonatal characteristics

About three fifth [251(59.48%)] of the neonates were females. One fifth of the neonates [87 (20.62%)] were born before 37 weeks of gestational age. Most of the neonates [254 (60.19%)]

**Table 1. Socio-demographic characteristics among postnatal mothers whose neonates admitted to neonatal intensive care unit of Debre Tabor General Hospital, Debre Tabor town, North Central Ethiopia, 2019 (n = 422).**

| Factor | N | % |
|---|---|---|
| **Residence** | | |
| Urban | 324 | 76.78 |
| Rural | 98 | 23.22 |
| **Maternal age (years)** | | |
| 16–20 | 29 | 6.87 |
| 20–34 | 322 | 76.30 |
| ≥34 | 71 | 16.82 |
| **Marital status** | | |
| Married | 406 | 96.21 |
| Other* | 16 | 3.79 |
| **Parity** | | |
| Primiparous | 223 | 52.84 |
| Multiparous | 199 | 47.16 |
| **Maternal educational status** | | |
| Unable to read and write | 76 | 18.01 |
| Primary education | 202 | 47.87 |
| Secondary education | 82 | 19.43 |
| College/university | 62 | 14.69 |
| **Husband's educational status** | | |
| Unable to read and write | 32 | 7.58 |
| Primary education | 127 | 30.09 |
| Secondary education | 148 | 35.07 |
| College/university | 115 | 27.25 |
| **Maternal occupation** | | |
| Civil servant | 216 | 51.18 |
| Merchant | 119 | 28.20 |
| House wife | 87 | 20.62 |
| **Average monthly income ($ US)** | | |
| **<37.5** | 139 | 32.94 |
| **≥37.5** | 283 | 67.06 |

* Other refers to divorced, widowed

were admitted to the hospital in the first 7 days of their postnatal age. One third of the neonates [140 (33.18%)] had low birth weight. At admission, neonates had several medical diagnoses of which hypothermia accounted for the highest percentage [295(69.91%)] followed by early onset neonatal sepsis [213(50.47%)] (**Table 3**).

## Maternal knowledge of neonatal uvula and traditional uvulectomy

More than half [230 (54.50%)] of the mothers mentioned none of the adverse effects of traditional neonatal uvulectomy. This quantitative finding can be supplemented by the qualitative evidence obtained from a 31 years old key informant mother who said "*All neonatal care providers in neonatal intensive care unit told me that disease condition of my kid was attributed to postuvulectomy infection. However, I strongly disagree with association of my neonatal illness to the procedure of traditional uvulectomy because the illness occurred one week after uvulectomy. If the illness had been attributed to uvulectomy, the kid could have been ill soon after the*

**Table 2. Obstetrics related factors among postnatal mothers whose neonates admitted at neonatal intensive care unitof Debre Tabor General Hospital, Debre Tabor town, North Central Ethiopia, 2019.**

| Factor | n | % |
|---|---|---|
| **ANC follow up (n = 422)** | | |
| Yes | 405 | 95.97 |
| No | 17 | 4.01 |
| **Number of ANC visits (n = 405)** | | |
| <4 times | 128 | 31.60 |
| ≥ 4 times | 277 | 68.40 |
| **Accompanied by spouse to ANC (n = 405)** | | |
| Yes | 125 | 30.86 |
| No | 280 | 69.14 |
| **Antenatal counseling of traditional neonatal uvulectomy (n = 405)** | | |
| Yes | 103 | 25.43 |
| No | 302 | 74.57 |
| **Antenatal couple counseling of traditional neonatal uvulectomy (n = 103)** | | |
| Yes | 44 | 42.72 |
| No | 59 | 57.28 |
| *What were you counseled? (n = 103)** | | |
| Adverse effects of traditional neonatal uvulectomy | 57 | 55.34 |
| Immediate modern health care seeking during perception of uvular swelling and elongation | 46 | 44.66 |
| The presence of modern medicine for elongated uvula | 33 | 32.04 |
| Benefits of uvula | 16 | 15.53 |
| **Ever had bad obstetrics history (n = 422)** | | |
| Yes | 75 | 17.77 |
| No | 347 | 82.23 |
| *If yes, which of the following? (n = 75)** | | |
| Neonatal death | 48 | 64.00 |
| Child death | 35 | 46.67 |
| Still birth | 32 | 42.67 |
| Abortion | 17 | 22.67 |
| IUFD | 9 | 12.00 |
| **Place of delivery (n = 422)** | | |
| Health institution | 286 | 67.77 |
| Home | 136 | 32.23 |
| **PNC visit (n = 422)** | | |
| Yes | 335 | 79.38 |
| No | 87 | 20.62 |
| *Number of PNC visits (n = 335)** | | |
| 1st PNC visit (Within 24 hours of birth) | 304 | 90.75 |
| 2nd PNC visit (1–7) days after birth | 46 | 13.73 |
| **Accompanied by spouse to PNC (n = 335)** | | |
| Yes | 317 | 94.63 |
| No | 18 | 5.37 |
| **Postnatal counseling about traditional neonatal uvulectomy(n = 335)** | | |
| Yes | 127 | 37.91 |
| No | 208 | 62.09 |
| **Postnatal couple counseling about traditional neonatal uvulectomy (n = 127)** | | |
| Yes | 48 | 37.80 |

(*Continued*)

**Table 2.** (Continued)

| Factor | n | % |
|---|---|---|
| No | 79 | 62.20 |
| *What were you counseled (n = 127) | | |
| Adverse effects of traditional neonatal uvulectomy | 85 | 66.93 |
| Immediate modern health care seeking during perception of uvular swelling and elongation | 54 | 42.52 |
| The presence of modern medicine for uvular swelling and elongation | 46 | 36.22 |
| Benefits of uvula | 22 | 17.32 |

*Multiple responses were given

**NB**: the third PNC visit wasn't considered as neonates are younger than this visit time.

procedure." Hence, these qualitative responses indicate maternal lack of awareness towards the so called incubation period, which is between the procedure of traditional uvulectomy and sepsis onset.

There were also 94 (22.27%) respondent mothers with prior experience of traditional neonatal uvulectomy for their elder children (Table 4).

## The burden of post uvulectomy admission rate

Out of 422 neonatal admissions, there were 67 (15.88%) postuvulectomy admissions (**Fig 1**). All the postuvulectomy neonatal admissions had at least one complication of which post uvulectomy sepsis accounted the highest burden [59 (88.06%)]. The mean neonatal age at uvulectomy was 5.42 days (SD = ±2.51). Majority [42 (62.69%)] of the uvulectomies were done in the first week of postnatal life **(Table 5)**.

Mothers suspect uvular swelling and elongation when their neonates develop different nonspecific symptoms, which are here considered as maternal perceived indicators of uvular swelling and elongation. Mothers of the uvulectomies [67 (15.88%)] were asked whether they had

**Table 3. Characteristics of the neonates admitted at neonatal intensive care unit of Debre Tabor General Hospital, Debre Tabor town, North Central Ethiopia, 2019.**

| Factor(n = 422) | Response | N | % |
|---|---|---|---|
| Sex | Male | 171 | 40.52 |
| | Female | 251 | 59.48 |
| Gestational age at birth | <37weeks | 87 | 20.62 |
| | ≥37weeks | 335 | 79.38 |
| Post natal age at admission (days) | < 7 | 254 | 60.19 |
| | ≥7 | 168 | 39.81 |
| Birth weight (grams) | <2500 | 140 | 33.18 |
| | ≥2500 | 282 | 66.82 |
| *Medical diagnosis @ admission | Hypothermia | 295 | 69.91 |
| | Early onset neonatal sepsis | 213 | 50.47 |
| | Late onset neonatal sepsis | 114 | 27.01 |
| | Prematurity | 87 | 20.62 |
| | Perinatal asphyxia | 75 | 17.77 |
| | Hypoglycemia | 62 | 14.69 |
| | Congenital defect | 58 | 13.74 |

*refers to the presence of multiple medical diagnoses for a neonate at admission

**Table 4. Knowledge of neonatal uvula and traditional uvulectomy among postnatal mothers whose neonates admitted at neonatal intensive care unit of Debre Tabor General Hospital, Debre Tabor town, North Central Ethiopia, 2019.**

| Factor | Response | n | % |
|---|---|---|---|
| Mentioned at least one benefit of uvula (n = 422) | Yes | 92 | 21.80 |
| | No | 330 | 78.20 |
| *The mentioned benefits of uvula **(n = 92)** | Preventing aspiration while swallowing breast milk | 77 | 83.70 |
| | Lubricating oropharyngeal mucosa | 18 | 19.57 |
| | Serving for language communication | 15 | 16.3o |
| | Boosting immunological function | 12 | 13.04 |
| | prevention of breast milk regurgitation through the neonatal nose | 9 | 9.78 |
| Mentioned at least one adverse effect of traditional neonatal uvulectomy (n = 422) | Yes | 192 | 45.50 |
| | No | 230 | 54.50 |
| *The mentioned adverse effects of traditional neonatal uvulectomy **(n = 192)** | Transmission of communicable infections (HIV, Hep B) | 146 | 76.04 |
| | Hemorrhage | 51 | 26.56 |
| | Tetanus | 39 | 20.31 |
| | Pharyngeal dryness | 25 | 13.02 |
| | Aspiration | 17 | 8.90 |
| | Pain | 10 | 5.21 |
| | Change in voice | 9 | 4.69 |
| | disturbance in sleep pattern | 7 | 3.65 |
| | Regurgitation of breast milk from the nostril | 7 | 3.65 |
| | Others | 9 | 4.69 |
| Prior history of traditional neonatal uvulectomy **(n = 422)** | Yes | 94 | 22.27 |
| | No | 328 | 77.73 |

* Multiple responses were given; others refer to tongue injury and neck swelling

perceived their own signals (indicators) of uvular swelling and elongation. And majority of the mothers reported that their neonates failed to breastfeed [16 (23.88%)] followed by fever as displayed by **Fig 2**.

Concerning maternal reasons of traditional neonatal uvulectomy, 53 (79.1%) mothers stated that elongated and swollen uvula can't be treated by modern medicine (**Fig 3**). Moreover, qualitatively, a 34 years of old mother said: *'When there is uvular swelling and elongation, contacting the traditional surgeon is the absolute medicine because, unless so, the elongated uvula becomes ruptured thereby causing inevitable neonatal death. For example, fearing this inevitable death, all my elder children had uvulectomy done during their neonatal lives after which they grew very well. There has been no modern treatment of elongated uvula since earlier times in our society'*

## Factors associated with traditional neonatal uvulectomy

From bivariable analysis, sex of the neonate, parity, place of delivery, antenatal couple counseling of traditional neonatal uvulectomy, postnatal couple counseling of traditional neonatal uvulectomy, mentioning at least one adverse effect of traditional neonatal uvulectomy, having history of traditional neonatal uvulectomy and history of bad obstetrics were significant factors. However, after adjusting for possible confounding effect, having male neonate

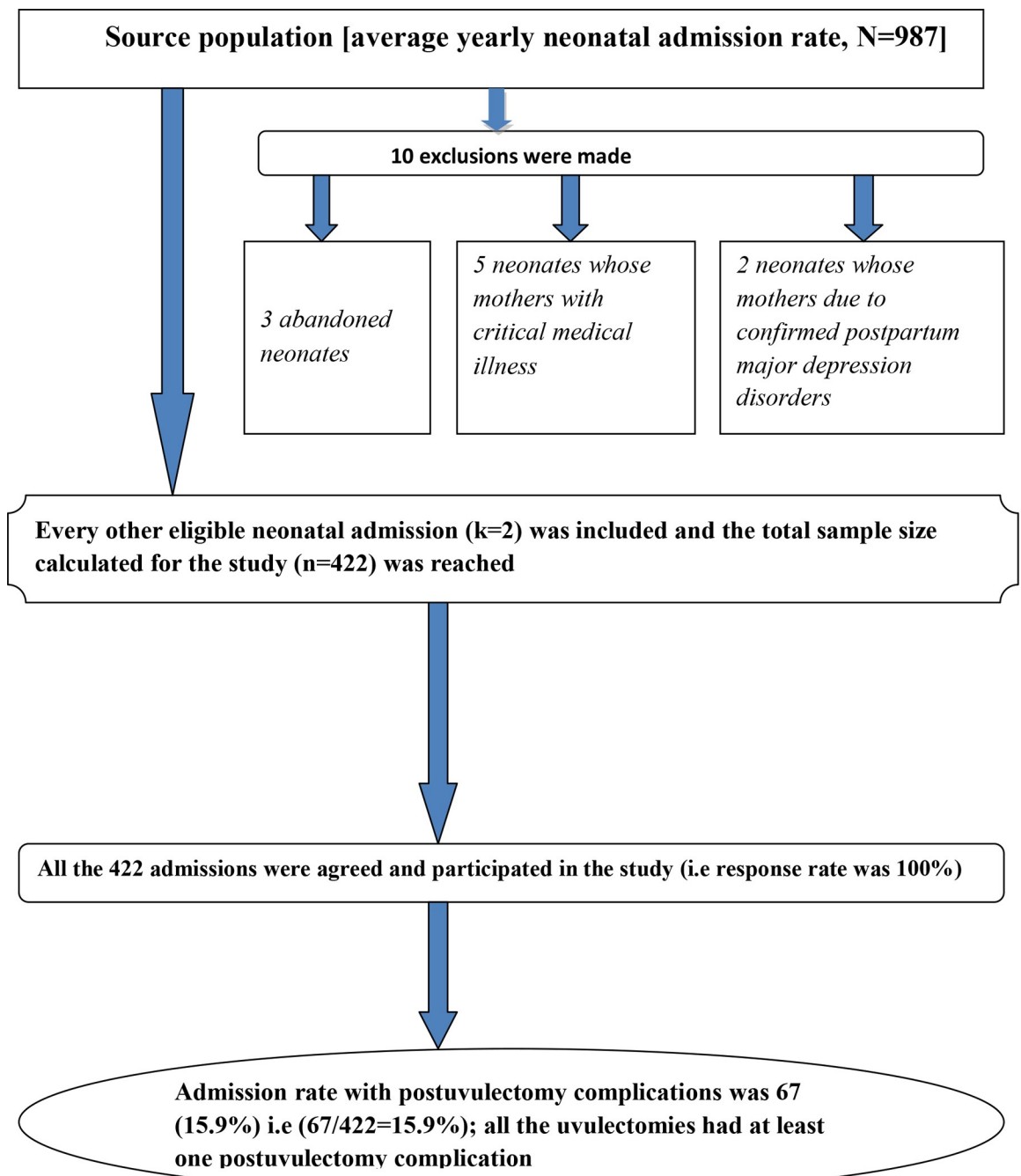

**Fig 1. A flow diagram illustrating inclusions, exclusions, admission percentages and complication rates of traditional neonatal uvulectomy among neonatal admissions at Debre Tabor General Hospital, South Gondar Zone, North Central Ethiopia, 2019.**

[AOR = 4.87; 95% CI: 1.10, 21.59], antenatal couple counseling of traditional neonatal uvulectomy [AOR = 0.053; 95% CI: 0.01, 0.35], home delivery [AOR = 6.02; 95% CI: 1.15, 31.61], postnatal couple counseling of traditional neonatal uvulectomy[AOR = 0.101; 95% CI: 0.02, 0.65], having history of traditional neonatal uvulectomy [AOR = 7.15; 95% CI: 1.18, 43.21] and mentioning at least one disadvantage of traditional neonatal uvulectomy [AOR = 0.068; 95% CI: 0.01, 0.44]were independent predictors of traditional neonatal uvulectomy.

**Table 5. Characteristics of postuvulectomy admissions to neonatal intensive care unit at Debre Tabor General Hospital, North Central Ethiopia, 2019.**

| Factor | Response | N | % |
|---|---|---|---|
| Neonatal age at uvulectomy (n = 67) | <7 days | 42 | 62.69 |
| | ≥ 7days | 25 | 37.31 |
| *Who did influence you to practice traditional neonatal uvulectomy? (n = 67) | Traditional uvulectomy practitioners | 27 | 40.30 |
| | Family | 23 | 34.33 |
| | Traditional birth attendants | 19 | 28.36 |
| | Friends | 14 | 20.90 |
| | Maternal own decision | 9 | 13.43 |
| What was the primary postuvulectomy complication at admission? (n = 67) | Sepsis | 59 | 88.06 |
| | Anemia | 37 | 55.22 |
| | Neck swelling | 11 | 16.42 |
| | Tongue and oropharyngeal injury | 7 | 10.45 |
| | Others* | 3 | 4.48 |

The odds of traditional uvulectomy among male neonates were 4.87 times higher as compared to female neonates [AOR = 4.87; 95% CI: 1.10, 21.59]. Neonates born to parents who were couple counseled of traditional neonatal uvulectomy during antenatal period were 94.7% less likely to be victim as compared to those neonates born to parents who weren't couple counseled [AOR = 0.053; 95% CI: 0.01, 0.35].

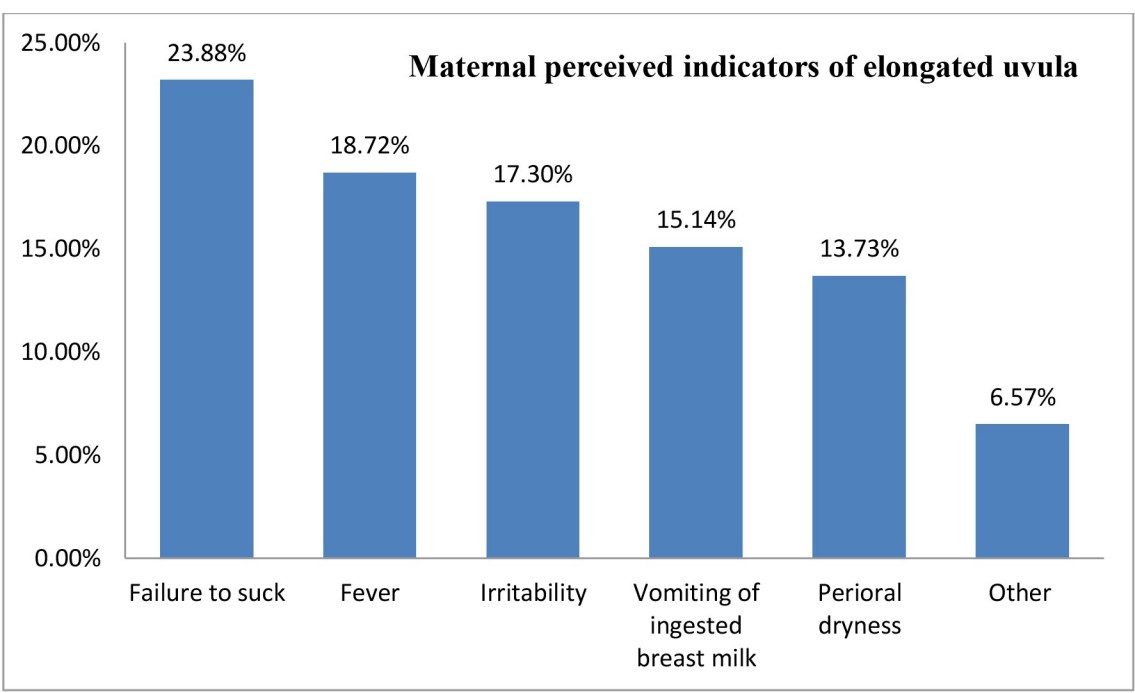

**Other** refers to "cough, skin rash, fontanel bulge, diarrhea and too sleepy neonates

**Fig 2. Perceived indicators for maternal suspicion of uvular swelling and elongation for their neonates, Debre Tabor General Hospital, South Gondar Zone, North Central Ethiopia, 2019.**

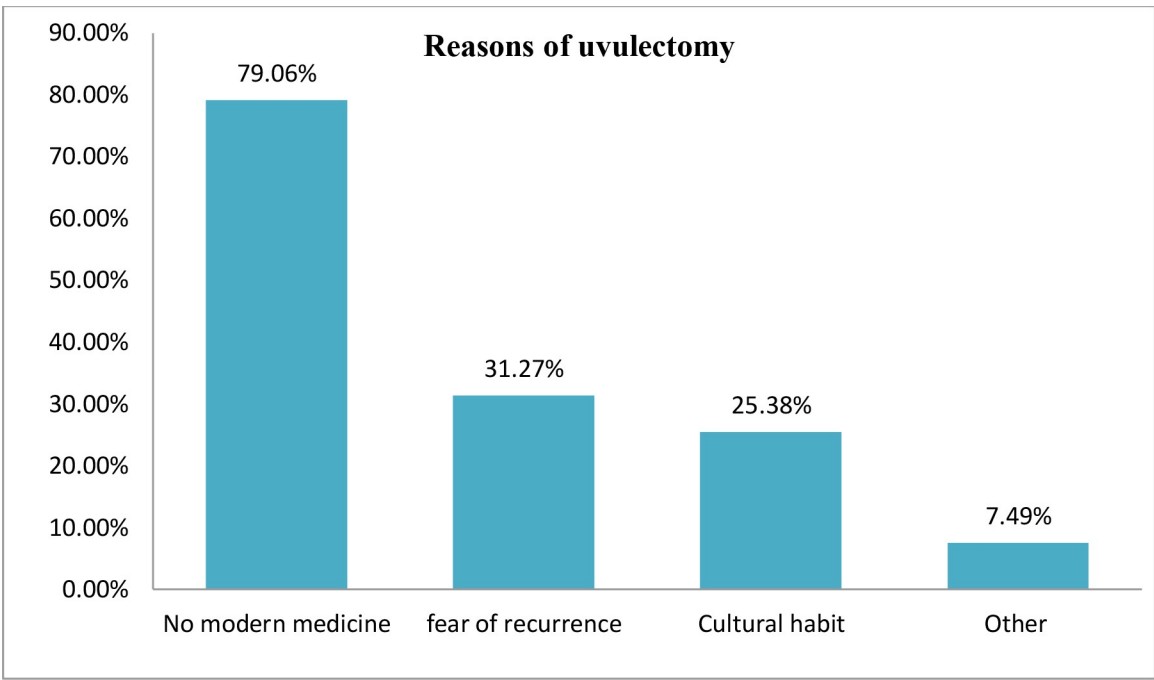

**Fig 3. Reasons of traditional neonatal uvulectomy among the post uvulectomy admissions at Debre Tabor General Hospital, South Gondar Zone, North Central Ethiopia, 2019.**

Home delivered neonates were 6.02 times more likely to have traditional uvulectomy when compared to those born at health institution [AOR = 6.02; 95% CI: 1.15, 31.61].It was supported by the qualitative data of a 25 years old mother who said: '*After I gave birth at my home, all the nearby people told me the essence of contacting traditional uvulectomy practitioners when my neonate becomes irritable, which is a warning signal of uvular swelling and elongation. Then, my neonate was done uvulectomy to get relieved of its repetitive spontaneous crying which I believed to be caused from uvular swelling and elongation*'.

The likelihood of traditional uvulectomy among neonates whose parents were couple counseled of traditional uvulectomy during postnatal visit was 89.9% lower than those whose parents weren't couple counseled [AOR = 0.101; 95% CI: 0.02, 0.65]. This finding was supported by a key informant mother who said: '*Just in front of our kid at post natal room, my husband and me were advised of the life threatening septic and hemorrhagic complications of traditional neonatal uvulectomy which we had never known before. It was heart touching to hear the advice in front of our kid. Since then, we promised never to experience traditional uvulectomy for our kid*'.

Neonates whose mothers mentioned at least one adverse effect of traditional neonatal uvulectomy were 93.2% less likely to be victim when compared to those whose mothers mentioned none of the adverse effects [AOR = 0.07; 95% CI: 0.01, 0.44] (Table 6). This is supported by the qualitative data obtained from a 23 year old mother who said: '*I observed when my neighbor's neonate was done uvulectomy. Some days after the procedure, the neonate developed severe illness manifested by bloody vomiting and difficulty of breathing. Just at that time, the parents were too much worried of the neonatal condition and hence we contacted the traditional practitioner who did the procedure, but he himself was very disturbed when he saw the neonate was vomiting blood. Ultimately, expecting no more solution from the uvulectomist, we brought the neonate to this hospital. The neonate got cured after it was oxygenated, given medications and*

**Table 6. Factors associated with traditional uvulectomy among neonatal admissions at neonatal intensive care unit of Debre Tabor General Hospital, North Central Ethiopia, 2019 (n = 422).**

| Factor | Traditional neonatal uvulectomy practice | | 95% CI | | |
|---|---|---|---|---|---|
| | Yes (%) | No (%) | Crude OR | Adjusted OR | P-value |
| **Sex of the neonate** (n = 422) | | | | | |
| Male | 57 (13.50) | 114 (27.01) | 12.10(5.94, 24.46) | 4.87(1.10, 21.59) | 0.025 |
| Female | 10 (2.37) | 241(57.11) | 1 | 1 | |
| **Parity** (n = 422) | | | | | |
| Primiparous | 45(10.66) | 178(42.18) | 2.03 (1.17, 3.53) | 1.42(.27, 7.37) | 0.089 |
| Multiparous | 22(5.21) | 177(41.94) | 1 | 1 | |
| **Antenatal couple counseling of traditional uvulectomy**(n = 103) | | | | | |
| Yes | 12 (11.65) | 32(31.07) | .030 (.016, .060) | .053 (.01, .35) | 0.001 |
| No | 55(53.40) | 4 (3.88) | 1 | 1 | |
| **Delivery place (n = 422)** | | | | | |
| Home | 58 (13.74) | 78 (18.48) | 8.41(4.68, 15.11) | 6.02 (1.15, 31.61) | 0.041 |
| Health institution | 9 (2.13) | 277 (65.64) | 1 | 1 | |
| **Postnatal couple counseling of traditional uvulectomy** (n = 127) | | | | | |
| Yes | 8 (6.30) | 40 (31.50) | .027 (.012, .06) | .101 (.02, .65) | 0.028 |
| No | 59(46.46) | 20(15.75) | 1 | 1 | |
| **Knowing at least one adverse effect of traditional uvulectomy** (n = 422) | | | | | |
| Yes | 4(0.95) | 188(44.55) | .056 (.02, .16) | .068 (.01, .44) | **0.004** |
| No | 63 (14.93) | 167(39.57) | 1 | 1 | |
| **Having history of traditional uvulectomy** (n = 422) | | | | | |
| Yes | 59(13.98) | 35(8.29) | 51.79(24.29, 110.43) | 7.15(1.18, 43.21) | **0.017** |
| No | 10 (2.37) | 318 (75.36) | 1 | 1 | |
| **History of bad obstetrics(n = 422)** | | | | | |
| Yes | 6(1.42) | 69 (16.35) | .12 (.065, .208) | .23 (.037, 1.423) | 0.473 |
| No | 61(14.45) | 286 (67.77) | 1 | 1 | |

*blood transfused. The neonatal care providers told us that the neonate suffered from hemorrhage during the procedure and also infection. Then, we became convinced and decided never to face traditional neonatal uvulectomy in our village again. The traditional practitioners received 200 ETB (Ethiopian Birr) per neonate thereby considering the malpractice as their source of income. They should be asked by law because they are endangering neonatal health by encouraging parents for uvulectomy rather than advising for modern medicine at hospital.'*

Although antenatal counseling of mothers alone about traditional uvulectomy wasn't a factor of significance, only a few neonates [7 (1.73%)]of the antenataly counseled mothers were done uvulectomy as compared to the larger proportion of uvulectomy cases [60(14.81%)] among mothers who weren't counseled. Similarly, there were fewer uvulectomy cases [9 (2.69%)] among the neonates of mothers who were postnatally counseled than the cases [58 (17.31%)] among those mothers who weren't counseled about traditional uvulectomy (Table 7).

## Discussion

This study addressed public health importance of traditional neonatal uvulectomy by showing its burden among neonatal admissions, associated factors and reasons in the study area. The burden of postuvulectomy admission was 67 (15.88%). Male sex, home delivery and prior history of traditional neonatal uvulectomy were significantly associated with increased odds of

**Table 7. Cross tabulation of antenatal and postnatal counseling of only mothers abut traditional neonatal uvulectomy with cases of uvulectomy at Debre Tabor General Hospital, South Gondar zone, North Central Ethiopia, 2019.**

| | | Uvulectomy done | | Total |
|---|---|---|---|---|
| | | Yes (%) | No (%) | |
| Antenatal counseling of traditional neonatal uvulectomy (n = 405) | Yes | 7 (1.73) | 96(23.70) | 103 (25.43) |
| | No | 60(14.81) | 242(59.76) | 302 (74.57) |
| Postnatal counseling of traditional neonatal uvulectomy (n = 335) | Yes | 9(2.69) | 118(35.22) | 127(37.91) |
| | No | 58(17.31) | 150(44.78) | 208 (62.09) |

traditional uvulectomy. On the other hand, knowing at least one adverse effect of traditional uvulectomy, antenatal and postnatal counseling of couples about traditional neonatal uvulectomy were significantly associated with decreased odds of traditional neonatal uvulectomy.

From this study, the burden of postuvulectomy admissions (15.9%) was consistent with a study in Niger (19.6%) [19]. However, it was lower than studies in Aksum (86.9%) [18] and Nigeria (86.1%) [28]; but higher than a Tanzanian study (1.0%) [11].The discrepancy may be due to differences in study setting, period, design and target population. Regarding complications of the uvulectomies, postuvulectomy sepsis [59 (88.1%)] was the leading complication at admission. This may be due to the use of unsterilized instruments on several neonates on the same session as stated by the key informants of this study. Most importantly, sepsis can be ensued by septicemia which is fatal [4, 6, 8, 12, 26, 28]. Therefore, preventive interventions like educating the community about case fatality of septic complications of uvulectomy should be targeted and exhaustively done.

Majority [42 (62.7%)] of the traditional uvulectomies were done in the first week of neonatal life, which is similar with a Nigerian study [26] showing 52.4% of the uvulectomies performed in the first week of life. Several studies showed that the first week of neonatal life is a critical time of morbidity and mortality mainly in developing countries [9, 12, 25, 30–32]. Thus, undergoing traditional uvulectomy during this time may accelerate early neonatal morbidity and mortality, which could fuel the national challenge of reducing neonatal mortality rate to (12/1000) by 2030 in Ethiopia. Therefore, exhaustive investment of different programmatic interventions that involve both governmental and nongovernmental organizations must be implemented in the community to save early neonatal lives from traditional uvulectomy. These organizations should ensure sustainability of their investment by enabling the community to safeguard neonates from traditional uvulectomy and other malpractices.

Globally, there is not a mention in the literature about indications of traditional uvulectomy [31]. However, in the study area, mothers advocate uvulectomy to heal their common thought of neonatal illness resulting from the suspicion of uvular swelling and elongation [10, 25]. Furthermore, the authors reached that maternal perception of "there is no modern medicine for treating elongated and swollen uvula" was the most mentioned and misconceived reason of contacting traditional practitioners.

The odds of traditional uvulectomy among male neonates were 4.87 times higher as compared to female neonates. This may be due to the fact that in the study area male sex is preferred and considered as a pride in the community. Thus, immediate traditional cares including traditional uvulectomy and prelacteal feeding are given if there is any perceived sign of illness to male neonates [29].Hence, for the context of this study, different stakeholders of neonatal health should educate the community by stressing on the key message of "declaring immediate traditional uvulectomy for ill male neonates means fueling masculine mortality from fatal complications of traditional uvulectomy thereby depriving community pride."

Regarding the counseling of traditional neonatal uvulectomy, 103 (63.0%) mothers were given antenatal counseling. Besides, 44 (42.7%) of the mothers were given the counseling together with their husbands. Neonates born to mothers who were couple counseled about traditional uvulectomy during antenatal period were 94.7% less likely to be victim of uvulectomy as compared to those born to mothers counseled alone. Besides, neonates born to mothers who were couple counseled of traditional neonatal uvulectomy in the postnatal periods were 89.9% less likely to be victim of the malpractice as compared to those born to mothers counseled alone. The assumption of couple counseling is that if husbands are present during counseling, mothers are more likely to comply with the counseling about natural benefits of uvula, the adverse effects of traditional uvulectomy and presence of modern treatment for neonatal illnesses attributed to perceived uvular swelling and elongation [12, 30].

The odds of traditional uvulectomy among home delivered neonates were 6.02 times higher as compared to those born at health institution. This could be due to the fact that, in Ethiopia, mothers who gave their birth at home don't usually attend postnatal care [30].Therefore, these mothers don't get postnatal couple counseling of traditional uvulectomy, which is a significant determinant of the malpractice as discussed in the aforementioned paragraph. Moreover, traditional birth attendants who assisted the home deliveries are thought to play their roles in encouraging traditional uvulectomy because they are influential in the community [8, 12, 18, 19, 21]. Therefore, maternal and neonatal health care providers should stress advocacy of institutional delivery to prevent different neonatal traditional practices including uvulectomy, all of which are continuations of home delivery [30].

Moreover, neonates born to mothers having prior history of traditional neonatal uvulectomy were 7.15 times more likely to experience the malpractice as compared to their counterparts. This may be due to the deeply rooted cultural advocacy of traditional neonatal uvulectomy for healing the rupture of elongated and swollen uvula thereby preventing inevitable neonatal death, which is a common misconception in the community [1, 4, 7, 8, 11, 18, 19].

Neonates whose mothers knew at least one adverse effect of traditional neonatal uvulectomy were 93.2% less likely to be victim of the malpractice when compared to those whose mothers knew none of the adverse effects. This could be due to the fact that if a mother is knowledgeable of the adverse effects, she becomes reserved of traditional neonatal uvulectomy to prevent her neonatal suffering [8]. Therefore, continuous training and retraining of community agents about the dangerous adverse effects (complications) of traditional neonatal uvulectomy should be instituted in the health care system of North Central Ethiopia.

Despite originality, it was a single center crossectional study limited to neonatal admissions at Debre Tabor General Hospital alone and hence it couldn't show the overall burden and admission outcome of traditional neonatal uvulectomy in the health care system. Mothers might have also failed to recall of their age and they could have given socially desirable answers to questions like their prior experience of traditional uvulectomy. Base line data for the study obtained from reports of Debre Tabor General Hospital and South Gondar Zone health departments might have also been influenced from reporting bias. Moreover, the qualitative responses were collected solely from mothers of the admitted neonates. Therefore, the authors recommend a multicenter cohort study to show wider picture of the burden of postuvulectomy admissions and their treatment outcomes in the health care system by involving traditional uvulectomists and health care providers as key informants.

## Conclusions and recommendation

The burden of traditional neonatal uvulectomy was high. Fortunately, its predictors namely sex of the neonate, antenatal and postnatal couple counseling of traditional neonatal

uvulectomy, home delivery, history of traditional neonatal uvulectomy and knowledge about the adverse effects of traditional neonatal uvulectomy are modifiable. Therefore, strong advocacy teams of neonatal health should be organized from the lowest administrative level (Gotes) to the highest level (Zone) to mobilize the community against traditional neonatal uvulectomy. The advocacy teams should involve different community groups mainly women health development armies, elders, religious fathers, health extension workers and health professionals. Moreover, criminal prosecutors should be engaged in the team to get legal concern of traditional neonatal uvulectomy as it elicits unnecessary complications that endanger neonatal lives. These advocacy teams should be strengthened and supported one another through their hierarchal referral linkages i.e (Gote ↔Kebele ↔Woreda↔ Zone).

Moreover, mothers and their husbands should be couple counseled about the life threatening complications of traditional neonatal uvulectomy during antenatal care. The couple counseling should also be an integral component of the postnatal care in the health care delivery system at South Gondar zone. Maternal and neonatal health care providers should also advocate the legal and public health interventions of eliminating this dangerous practice. Besides, designing strategies to enhance community health care seeking behavior during parental suspicion of neonatal illness attributed to uvular swelling and elongation is another important method of preventing traditional neonatal uvulectomy.

## Supporting information

**S1 File. (Questionnaire): A structured questionnaire used for interviewing selected mothers about traditional neonatal uvulectomy, North Central Ethiopia, DTGH, 2019.**
(DOCX)

**S2 File. (Interview guide): A structured interview guide used for interviewing selected key informant mothers about their perception and experience of traditional neonatal uvulectomy, DTGH, North Central Ethiopia, 2019.**
(DOCX)

## Acknowledgments

The author acknowledged the director of Debre Tabor General Hospital, data collectors, supervisors and data entry operators. The author is also deeply indebted to the Institutional Health Research Ethics Review Committee (IHRERC) of Debre Tabor University for working on the ethical perspectives of the proposal and letting do this study. Last but not least, the respondents deserve the authors' sincerest thanks for their kind responses.

## Author Contributions

**Conceptualization:** Wubet Alebachew Bayih.

**Formal analysis:** Wubet Alebachew Bayih.

**Funding acquisition:** Wubet Alebachew Bayih.

**Investigation:** Wubet Alebachew Bayih.

**Methodology:** Wubet Alebachew Bayih.

**Resources:** Wubet Alebachew Bayih.

**Software:** Wubet Alebachew Bayih, Biniam Minuye Birhan, Abebaw Yeshambel Alemu.

**Supervision:** Wubet Alebachew Bayih, Biniam Minuye Birhan, Abebaw Yeshambel Alemu.

**Validation:** Wubet Alebachew Bayih, Biniam Minuye Birhan, Abebaw Yeshambel Alemu.

**Visualization:** Wubet Alebachew Bayih.

**Writing – original draft:** Wubet Alebachew Bayih, Biniam Minuye Birhan, Abebaw Yeshambel Alemu.

**Writing – review & editing:** Wubet Alebachew Bayih, Biniam Minuye Birhan.

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
