## [Decision Letter · Decision Letter 0]

2 Mar 2020

PONE-D-20-01248

The burden of traditional neonatal uvulectomy among admissions to neonatal intensive care units, North Central Ethiopia, 2019: A triangulated crossectional study

PLOS ONE

Dear Mr. Alebachew,

Thank you for submitting your manuscript to PLOS ONE. After careful consideration, we feel that it has merit but does not fully meet PLOS ONE’s publication criteria as it currently stands. Therefore, we invite you to submit a revised version of the manuscript that addresses the points raised during the review process.

We would appreciate receiving your revised manuscript by Apr 16 2020 11:59PM. To enhance the reproducibility of your results, we recommend that if applicable you deposit your laboratory protocols in protocols.io, where a protocol can be assigned its own identifier (DOI) such that it can be cited independently in the future. For instructions see: http://journals.plos.org/plosone/s/submission-guidelines#loc-laboratory-protocols

We look forward to receiving your revised manuscript.

Kind regards,

Ju Lee Oei

Academic Editor

PLOS ONE

Additional Editor Comments (if provided):

Thank you for the submission of this very interesting paper. Please address the comments from the reviewers, especially reviewer #2. The paper will also need substantial editing by a native English speaker prior to resubmission.

Journal Requirements:

2. Please include additional information regarding the survey or questionnaire used in the study and ensure that you have provided sufficient details that others could replicate the analyses. For instance, if you developed a questionnaire as part of this study and it is not under a copyright more restrictive than CC-BY, please include a copy, in both the original language and English, as Supporting Information.  If the original language is written in non-Latin characters, for example Amharic, Chinese, or Korean, please use a file format that ensures these characters are visible.

3. Please state whether you validated the questionnaire prior to testing on study participants. Please provide details regarding the validation group within the methods section.

4. Please include a copy of the interview guide used in the study, in both the original language and English, as Supporting Information, or include a citation if it has been published previously.

5. Please amend your current ethics statement to address the following concerns: Please explain why was written consent was not obtained, how you recorded/documented participant consent, and if the ethics committees/IRBs approved this consent procedure.

Reviewers' comments:

Reviewer's Responses to Questions

**Comments to the Author**

1. Is the manuscript technically sound, and do the data support the conclusions?

Reviewer #1: Yes

Reviewer #2: No

2. Has the statistical analysis been performed appropriately and rigorously? 

Reviewer #1: I Don't Know

Reviewer #2: No

3. Have the authors made all data underlying the findings in their manuscript fully available?

Reviewer #1: Yes

Reviewer #2: No

4. Is the manuscript presented in an intelligible fashion and written in standard English?

Reviewer #1: Yes

Reviewer #2: No

5. Review Comments to the Author

Reviewer #1: minor changes and revision to highlight the lack of understanding and awareness and the role education plays in improving neonatal mortality from traditional unscientific practices. similar traditional practices about care of umbilical cord leading to sepsis and neonatal tetanus have been decreased with education provided to the community nurses, midwives and pregnant women.

Reviewer #2: Dear authors,

Thank you for submitting this very interesting paper. Although it seems like a very important issue to address, the paper needs major revisions before it can be suitable for publication.

Please find my comments attached.

Yours sincerely,

6. PLOS authors have the option to publish the peer review history of their article (what does this mean?). If published, this will include your full peer review and any attached files.

Reviewer #1: No

Reviewer #2: No

---

## [Author Response · Author response to Decision Letter 0]

24 Apr 2020

Response letter

 Dear academic Editor (Ju Lee Oei)

After going through the entire manuscript, you forwarded your constructive comments which we missed to touch. Therefore, we are glad enough to express our sincerest thanks for your constructive editorial comments that could help improve novelty of our effort.

Editor’s comment: Thank you for the submission of this very interesting paper. Please address the comments from the reviewers, especially reviewer #2. The paper will also need substantial editing by a native English speaker prior to resubmission.

Authors’ response: Sure! We have tried our best to address the comments from the reviewers, especially reviewer #2 as detailed in the point by point responses stated in the following subsequent author responses for each of the reviewer’s comments. Moreover, from repeated proof-reading of the manuscript, we found several grammatical errors, interlinings, police titles, punctuation errors, wordings and spelling errors. Therefore, finding our colleague who has Master of Arts in English, we have did our best to thoroughly copyedit the manuscript for English language usage. These editorial changes are found throughout the revised version manuscript.

Editor’s comment: Please ensure that your manuscript meets PLOS ONE's style requirements, including those for file naming. The PLOS ONE style templates can be found at http://www.plosone.org/attachments/PLOSOne_formatting_sample_main_body.pdf and http://www.plosone.org/attachments/PLOSOne_formatting_sample_title_authors_affiliations.pdf

Authors’ response:Yes indeed, accessing the PLOS ONE style templates from the given links, our manuscript has been made to meet PLOS ONE's style requirements, including those for file naming. These changes were made to meet PLOS ONE's style requirements and found throughout the revised version manuscript.

Editor’s comment: Please include additional information regarding the survey or questionnaire used in the study and ensure that you have provided sufficient details that others could replicate the analyses. For instance, if you developed a questionnaire as part of this study and it is not under a copyright more restrictive than CC-BY, please include a copy, in both the original language and English, as Supporting Information. If the original language is written in non-Latin characters, for example Amharic, Chinese, or Korean, please use a file format that ensures these characters are visible.

 Authors’ response: The most invaluable comment it is! Hence, we have included additional information regarding the questionnaire used in the study and ensured the presence of sufficient details so that others could replicate the analyses. The questionnaire was developed as part of this study and has been included as ‘Supporting Information’ both in the original language (Amharic) and English. The original language is written in Amharic, having non-Latin characters, and hence a file format that ensures visibility of these characters is used. We are working on copy right of the questionnaire developed as part of this study.

Amendment to the original manuscript can be noticed from the yellow highlighted text on page 5 of the revised version manuscript. Besides, the questionnaire file legend is located at the end of the manuscript named as ‘additional file 1’ and uploaded as ‘supporting information’. 

Editor’s comment: Please state whether you validated the questionnaire prior to testing on study participants. Please provide details regarding the validation group within the methods section.

Authors’ response: Exactly! As you commented, the questionnaire was validated on study participants before the actual data collection through pretesting on 21 eligible mother-baby pairs (5% of sample size) at Debre Tabor General Hospital just 2 weeks prior to the study. Based on the pretesting, clarity of questions, wordings, sequence of questions and reaction of the respondents to the questions was evaluated after which modifications were made to the tool. 

Improvement to the original manuscript can be noticed from the methods section, subsection of data Quality control as shown by the yellow highlighted text on page 6 of the revised version manuscript.

Editor’s comment: Please include a copy of the interview guide used in the study, both in the original and English languages, as Supporting Information, or include a citation if it has been published previously.

Authors’ response: We strongly agreed with relevance of including the interview guide used in the study, in both the original language (Amharic) and English, as ‘Supporting Information’. Thus, both language versions (Amharic and English) of the interview guide used for the study have been included as ‘supporting information’ as shown by the yellow highlighted text on page 5 of the revised version manuscript. Besides, the interview guide file legend is located at the end of the manuscript named as ‘additional file 2’ and uploaded as supporting information.

 Editor’s comment: Please amend your current ethics statement to address the following concerns: Please explain why was written consent was not obtained, how you recorded/documented participant consent, and if the ethics committees/IRBs approved this consent procedure.

Authors’ response: Undoubtedly! This comment is helpful to secure ethical perspective of the study! Hence, authors reached that obtaining only informed voluntary verbal consent was enough for ethical approval by the ethics committee due to the following reasons: I) regarding women’s educational status, the authors had prior data indicating that nearly half (48%) of the women in Ethiopia didn’t have the ability to read and write [9]. II) The study was an interviewer based crossectional study aimed for the direct beneficence of mothers in improving neonatal health through boosting their awareness towards the adverse health impact of traditional neonatal uvulectomy III) The study didn’t also involve any measurement that could bring physical harm to the mothers and their neonates. IV) Each respondent’s informed verbal voluntary consent was marked as ‘√’ in the consent form just in front of the participant and its copy was given to the participant. Besides, the consent form was recorded in the cover page of hardcopy of the questionnaire and interview guide where it can stay there as long as possible. Therefore, taking all the aforementioned preconditions into consideration, the Institutional Health Research Ethics Review Committee (IHRERC) of Debre Tabor University assured ethical approval of the study. 

The detailed modification has been included in the methods section, ethical approval and consent to participate subsection of the revised version manuscript, page 7, as shown by the yellow highlighted text.

Dear Reviewer #1 

After going through the entire manuscript, you forwarded your constructive comment which we missed to touch. Therefore, we are glad enough to express our sincerest thanks for your in-depth review and comments that could help improve novelty of our efforts.

Reviewer Comment: Minor changes and revision to highlight the lack of understanding and awareness and the role education plays in improving neonatal mortality from traditional unscientific practices. Similar traditional practices about the care of umbilical cord leading to sepsis and neonatal tetanus have been decreased with education provided to the community nurses, midwives and pregnant women.

Authors’ response: Definitely! From exhaustive teaching of the community, there has been a global improvement of neonatal mortality from traditional practices about the care of umbilical cord leading to sepsis and neonatal tetanus. In our study area, the community mistakenly attributes nearly all neonatal illnesses to uvular swelling and elongation. Thus, ill neonates are often subjected to traditional uvulectomy for misconceived better cure [10, 25]. Therefore, educating the community sustainably about the harmful effects of traditional uvulectomy is thought to bring behavioral change in the study area so that it could be possible to reduce neonatal mortality from traditional uvulectomy [12, 29, 30].

The amendment is located on page 4 of the revised version manuscript as shown by the yellow highlighted text of the background section”.

Dear reviewer 2 

After going through the entire manuscript, you forwarded your constructive comments which we missed to touch. Therefore, we are glad enough to express our sincerest thanks for your in-depth review and comments that could help improve novelty of our effort.

Reviewer Comment: Throughout the article traditional uvulectomy is referred to as “… the malpractice” which is not objective, so would suggest to change to ‘traditional uvulectomy’, unless it flows naturally from the sentence before like the last line of the introduction (“Nonetheless, there is no current regional and even national data about the proportion, associated factors and reasons of this malpractice.”).

Authors’ response: Yes, as you said, the expression “this malpractice” has been replaced by “traditional uvulectomy” unless it flows naturally from the sentence before.

The amendment is located throughout the revised version manuscript as shown by the yellow highlighted text renamed as “traditional uvulectomy”.

Reviewer Comment: Reference 10 and 25 should be mentioned in the text, not as reference. The references in general should be thoroughly revised as they do not meet the APA standards. 

Authors’ response: We acknowledge incorporating the reviewer’s comment to this document. However, to the authors’ understanding of PLOS ONE journal requirement, Vancouver citation and referencing is allowed. Besides, we have ensured that references 10 and 25 have been mentioned in the reference list whilst describing these references are ‘unpublished’. 

The improvement is located in the background and methods section, pages 23 & 24 of the revised version manuscript as shown by the yellow highlighted text. 

Reviewer Comment: My most important comment/concern is the presentation of the study results. For me it is unclear how many uvulectomies were done, if there were potentially uvulectomies missed (bias) and how the admission percentages were calculated. From the abstract and methods it seems that 422 women were admitted to hospital, but then in the discussion there is a 15% admission rate mentioned? A flow diagram with inclusions, exclusions and complication rates would be helpful. 

Authors’ response: Very important comment it is! At first, 10 mothers comprising 2.4% of the sample size (422) were excluded. These mothers were (3 mothers of abandoned neonates because there was no other source of subjective data for these neonates, 5 mothers with critical medical illness and 2 mothers due to confirmed postpartum major depression disorders as these mothers were not mentally and physically capable of being interviewed). Then, every other (K=2) eligible mother-neonate pairs admitted to neonatal intensive care unit during the study was included to the study to reach our calculated sample size which is 422 admissions. Among these admissions, there were 67 (15.9%) uvulectomies. All the postuvulectomy admissions had at least one complication and post uvulectomy sepsis [59 (88.1%)] was the commonest complication (Figure 1). 

Improvement has been included to the revised version manuscript as shown by the yellow highlighted text of the methods section, subsection of ‘study design and participant characteristics’ (pages 4&5) and results section, subsection of the burden of post uvulectomy admission rate (page 13).

Reviewer Comment: “Male sex, home delivery and prior history of traditional uvulectomy were significantly associated with increased odds of traditional neonatal uvulectomy.” Is this associated with the uvulectomy itself or with the complications of uvulectomy? 

Authors’ response: The authors would like to ask great excuse for several statements whose central message hard to catch. Thus, taking the comment into account, the aforementioned factors (Male sex, home delivery and prior history of traditional uvulectomy) were associated with the uvulectomy itself than with the complications of uvulectomy. 

Easier statement has replaced the original write up as can be seen from the revised version manuscript on page 18, discussion section of the revised version manuscript as shown by the yellow highlighted text.

Reviewer Comment: “Mixed type (quantitative supplemented with qualitative) hospital based cross sectional study was conducted. Phenomenological study design was employed for the qualitative part. All postnatal mothers whose age ≥ 18 years and lived at least six months in the study area prior to the study and visiting Neonatal Intensive Care Unit (NICU) during the study were eligible. However, 3 abandoned neonates (those neonates left in NICU without mothers), 5 mothers with critical illness or any difficulty of talking/listening, 2 mothers having psychiatric disorders and known medical problems were excluded since they were not mentally and physically capable of being interviewed. Non volunteer mothers were also excluded.”This doesn’t sound free of bias, it seems that there might be a lot of cases missed and that only the potential ‘tip of the iceberg’ has been investigated. 

Authors’ response: Really! We found this comment with greatest value! Hence, elaboration has been emphasized to inform readers about the presence of only 10 admissions that were excluded of the study (3 mothers of abandoned neonates, 5 mothers with critical medical illness and 2 mothers due to confirmed postpartum major depression disorders). Otherwise, there was not any more exclusion criteria considered. The statement “All postnatal mothers whose age ≥ 18 years and lived at least six months…” is editorial problem because there was not age and residence related exclusion of study subjects. Moreover, there was no none volunteer mother for the study because every approached mother was volunteer and participated in the study which can be witnessed from 100% response rate. Therefore, considering the aforementioned corrections, the authors believe that introduction of bias into the study has been minimized and lots of cases were allowed to be included rather than letting investigation of only the potential ‘tip of the iceberg’.

Amendment to the revised version manuscript has been incorporated as displayed by the yellow highlighted text in the methods section, subsection of study design and participant characteristics, pages 4 & 5. 

Reviewer Comment: The sample size calculations are based on quite a few assumptions and the sample size of the qualitative study seems very small (n=8).

Authors’ response: First of all, we acknowledge your concern to know our assumptions of calculating sample size for the qualitative study. The sample size (n=8) for the qualitative part of the study was not predetermined as to the quantitative one. Rather, it was reached when repetitive responses were recorded by the principal investigator and it happened after interviewing 8 mothers.

Reviewer Comment:“Regarding the counseling of traditional neonatal uvulectomy, 103 (63.0%) mothers were given antenatal counseling. Besides, 44 (42.7%) of the mothers were given the counseling together with their husbands.”This should be addressed in the discussion. What is the assumption here? That if husbands are present they are more or less likely to comply with counselling? 

Authors’ response: Exactly! We the authors are much comforted with public health importance of addressing this comment in the discussion. The assumption of antenatal couple counseling of traditional neonatal uvulectomy is that if husbands are present during antenatal counseling, mothers are more likely to comply with the counseling i.e likelihood of practicing uvulectomy decreases. Based on the raised comment, the following statements are included in the discussion.

“Regarding the counseling of traditional neonatal uvulectomy, 103 (63.0%) mothers were given antenatal counseling. Besides, 44 (42.7%) of the mothers were given the counseling together with their husbands. Neonates born to mothers who were couple counseled about traditional uvulectomy during antenatal period were 94.7% less likely to be victim of uvulectomy as compared to those born to mothers counseled alone. Besides, neonates born to mothers who were couple counseled of traditional neonatal uvulectomy in the postnatal periods were 89.9% less likely to be victim of the malpractice as compared to those born to mothers counseled alone. The assumption of couple counseling is that if husbands are present during counseling, mothers are more likely to comply with the counseling about natural benefits of uvula, the adverse effects of traditional uvulectomy and presence of modern treatment for neonatal illnesses attributed to perceived uvular swelling and elongation [12, 30].”

Improvement has been included to the revised version manuscript as shown by the yellow highlighted text of the ‘discussion section, page 19.

Reviewer Comment: “Neonates born to parents who were couple counseled of traditional neonatal uvulectomy during antenatal period were 94.7% less likely to be victim as compared to those neonates born to parents who weren’t couple counseled [AOR= 0.053; 95% CI: 0.01, 0.35].” This is a very important conclusion of the paper that should be stressed more (i.e. included in abstract and conclusion). On which data is this conclusion based?

Authors’ response: Yes indeed! We found this comment with paramount significance of programmatic implication for intervening on the problem. The aforementioned conclusion was reached from the analyzed data on 103 mothers who were antenatally counseled of traditional neonatal uvulectomy. Based on a 2 by 2 dummy table, the analyzed data were: proportion of uvulectomies among neonates whose parents were antenatally couple counseled of traditional neonatal uvulectomy [12 (11.65%)], proportion of uvulectomies among not antenatally couple counseled of traditional neonatal uvulectomy [55(53.40%)], proportion of no uvulectomies among antenatally couple counseled of traditional neonatal uvulectomy [32(31.07%)] and proportion of no uvulectomies among not antenatally couple counseled of traditional neonatal uvulectomy [4(3.88%)]. Based on the comment, this finding has just been emphasized both in the abstract and conclusion sections of the revised version manuscript as shown by the yellow highlighted text, page 3 & 21.

Reviewer Comment:‘After I delivered at home, all the men and women who helped me during the birth were dealing with the essence of contacting traditional uvulectomy practitioners if my kid becomes irritable despite good breastfeeding. This is because nowadays elongated uvula is chiefly characterized by irritability rather than decreased breastfeeding. Then, I experienced traditional uvulectomy when I was in trouble of the kid’s spontaneous crying despite its successful breastfeeding’ – what does this mean? The statements of the qualitative responses are hard to interpret in general. Could there be a better way of including this information? Most of the statements are very valuable, but they are rather long. I would suggest a major revision of the paper to make this information more easily digestible.

Authors’ response: Great thanks! Having this comment, we thoroughly read every sentence based on which we tried to make it specific and concise for more easily digestible transfer of the required information. The above statements are restated as below.

‘After I gave birth at my home, all the nearby people told me the essence of contacting traditional uvulectomy practitioners when my neonate becomes irritable, which is a warning signal of uvular swelling and elongation. Then, my neonate was done uvulectomy to get relieved of its repetitive spontaneous crying which I believed to be caused from uvular swelling and elongation’

Improvement has been included to the revised version manuscript as shown by the yellow highlighted text of the results section, subsection of ‘Factors associated with traditional neonatal uvulectomy’, page 15.

Reviewer comment:Table 5 says “Neonatal age at uvulectomy n=67”; does this mean that only 67 of the 422 mothers answered this question? 

Authors’ response: Definitely! Out of the 422 mothers, there were only 67 mothers whose neonates were done uvulectomy and hence“Neonatal age at uvulectomy n=67” means only 67 of the 422 mothers answered this question. 

Reviewer comment: Forty percent of information about traditional uvulectomy comes from traditional uvulectomy surgeons (!). Is there a way to counsel them and if so, could this be discussed in the discussion?

Authors’ response: Acknowledging the reviewer’s comment about the way to counsel traditional uvulectomy surgeons and discussion of the counseling message, there was no any possible means of accessing the traditional practitioners due to hospital based nature of the study, and it has been explained as limitation of the study.

Improvement has been incorporated within discussion section of the revised version manuscript, pages 20 & 21 as shown by the yellow highlighted text.

Reviewer Comment:The likelihood of traditional uvulectomy among neonates whose mothers and fathers received counseling of traditional uvulectomy during postnatal visit was 89.9% lower than those whose parents weren’t couple counseled [AOR= 0.101; 95% CI: 0.02, 0.65]. This finding was supported by a key informant who said: ‘Just in front of our kid at post natal room, my husband and me were advised of the life threatening septic and hemorrhagic complications of traditional neonatal uvulectomy which we had never known before. It was heart touching to hear the advice in front of our kid. Since then, we promised never to experience traditional uvulectomy for our kid’. This seems to contradict with the information given in Table 2. In Table 2, 63% received neonatal counseling, yet they all had traditional uvulectomy. 

Authors’ response: Acknowledging your concern about the role of neonatal counseling relative to traditional uvulectomies, the statistics 103(63%) refers to those mothers who received antenatal counseling about traditional uvulectomy and only 7(6.5%) of them experienced uvulectomy for their neonates as can be seen from the cross-tabulation in table 7, page 18 of the revised version manuscript. Moreover, regarding postnatal couple counseling, which is the concern of this comment, only 8 (6.30%) of the neonates whose parents couple counseled of uvulectomy, were done uvulectomy as displayed in table 6, page 17 of the revised version manuscript. Therefore, as to our understanding, the two sentences supplement each other. 

Reviewer Comment: With all comments taken into account, the discussion should be revised and resubmitted for further review. Most importantly, the results and consequences of current practice should be discussed. There are many potential limitations including several biases (selection bias, inclusion bias, reporting bias,) that should be discussed. 

Authors’ response: We are really convinced of the reviewer’s comments regarding discussion. Thus, based on the comment, current results and consequences of traditional neonatal uvulectomy are discussed. Moreover, clear briefing of programmatic implications for significant factors has been incorporated in the revised version manuscript. The potential limitations including several biases (selection bias, inclusion bias, reporting bias,) have also been discussed.

The detailed modification has been made to the original manuscript as can be seen from the yellow highlighted text, pages 18- 21 of the revised version manuscript.

Other minor comments:

Reviewer Comment (Introduction): “There are divergent views to the reason as well as its overall benefit in these countries.” This statement seems taken out of the air without references or examples. 

Authors’ response: We found this comment with greatest relevance because it helps readers understand where this message sourced from. Hence, referring to the endnote library, the appropriate reference has been given to the aforementioned statement as shown by the yellow highlighted reference within the square bracket on page 3 of the revised version manuscript.

Reviewer comment (Table 1): I assume the 4th line under husband’s educational status should say college/university instead of secondary education twice.

Authors’ response: Absolutely right! The 4th line under husband’s educational status should say college/university instead of secondary education twice. Hence, the second ‘secondary education’ has been replaced by college/university as shown by the yellow highlighted text on page 9 of the revised version manuscript.

Reviewer Comment (Results): “Neonates’ failure to breast feed 16 (23.9%) was the most reported indicator of elongated uvula (Figure 2).” What do the authors mean by this? 

Authors’ response: Certainly! This comment could have been explained in the original manuscript. Hence, the following correction has been included to the revised version manuscript as shown by the yellow highlighted text, pages 14.

“In the study area, the community mistakenly attributes nearly all neonatal illnesses to uvular swelling and elongation [10, 25]. Thus, ill neonates are often subjected to traditional uvulectomy for misconceived cure. Mothers suspect uvular swelling and elongation when their neonates develop different nonspecific symptoms, which are here considered as maternal perceived indicators of uvular swelling and elongation. Mothers of the uvulectomies [67 (15.9%)] were asked whether they had perceived their own signals (indicators) of uvular swelling and/ elongation. And majority of the mothers reported that their neonates failed to breastfeed [16 (23.9%)] followed by fever (18.7%), irritability (17.3%), vomiting (15.1%), perioral dryness (13.7%) and others (6.5%) as displayed by Figure 2. 

Reviewer Comment (Table 6): I would suggest removing the significance from the bottom and just adding an extra column with p-values. 

Authors’ response: Removing the significance from the bottom of the regression table, we have just added an extra column with p-values as shown by the yellow highlighted column of the regression table, page 17 of the revised version manuscript.

Reviewer comment (Discussion): tried to address … suggest to change to ‘addressed the’ 

Authors’ response: Certainly! According to the given suggestion, ‘tried to address’ has been changed to ‘addressed the’, as can be noticed from the yellow highlighted text in the discussion section, first paragraph, page 18 of the revised version manuscript.

Reviewer comment (Figures): The Figures are not very informative to me. Please reconsider the information provided.

Authors’ response: Definitely! Based on the comment, the figures have been reconsidered to make them informative. Besides, informative textual explanation has been given to the figures as indicated by the yellow highlighted text of the results section, subsection of the burden of post uvulectomy admission rate, pages 14 & 15.

Textual suggestions

Reviewer Comment: In general, the quality of the English is poor; I would suggest a language revision by a preferably native English speaker. 

Authors’ response: Undoubtedly! From repeated proof-reading of the manuscript, we found several grammatical errors, interlinings, police titles, punctuation errors, wordings and spelling errors. Therefore, finding our colleague who has Masters of Arts in English, we have tried our best to thoroughly copyedit the manuscript for English language usage. These changes are found throughout the revised version manuscript.

Reviewer Comment (disadvantages) suggest to remove

Authors’ response: The authors strongly agreed with the suggested removal of “disadvantages” from throughout the manuscript and its replacement with ‘adverse effects’.

Improvement can be appreciated from throughout the revised version manuscript as can be seen by the yellow highlighted ‘adverse effects’ . 

Reviewer Comment:Top of page 5 ‘literatures” suggest to change to ‘studies’

Authors’ response: Very important comment it is! Thus, ‘literatures’ has been changed to ‘studies’ as per the given comment. 

Amendment can be appreciated from methods section, measurement and data collection procedure subsection, page 5 of the revised version manuscript as shown by the yellow highlighted text. 

Reviewer Comment: ? DTGH – change to ‘the hospital’

Authors’ response: we strongly agree with the raised comment and hence the abbreviation ‘DTGH’ has been changed to ‘Debre Tabor General Hospital’ as shown by the yellow highlighted text at various page numbers of the revised version manuscript.

Reviewer Comment Punctuation should be consistently checked; consistency of capitals (i.e. Table 1, Figure 1, etc.) and (n=xx, xx%) in correct form.

Authors’ response: We have no doubt with this comment! Hence, consistency of capitals and punctuation has been checked throughout the revised version manuscript as can be seen from the yellow highlighted Table and Figure titles.

---

## [Decision Letter · Decision Letter 1]

5 May 2020

PONE-D-20-01248R1

The burden of traditional neonatal uvulectomy among admissions to neonatal intensive care units, North Central Ethiopia, 2019: A triangulated crossectional study

PLOS ONE

Dear Mr. Alebachew,

Thank you for submitting your manuscript to PLOS ONE. After careful consideration, we feel that it has merit but does not fully meet PLOS ONE’s publication criteria as it currently stands. Therefore, we invite you to submit a revised version of the manuscript that addresses the points raised during the review process.

The manuscript is much improved. However, please edit it for minor grammatical details and presentation. For example, two decimal points are not needed in many of the results.

We would appreciate receiving your revised manuscript by Jun 19 2020 11:59PM. To enhance the reproducibility of your results, we recommend that if applicable you deposit your laboratory protocols in protocols.io, where a protocol can be assigned its own identifier (DOI) such that it can be cited independently in the future. For instructions see: http://journals.plos.org/plosone/s/submission-guidelines#loc-laboratory-protocols

We look forward to receiving your revised manuscript.

Kind regards,

Ju Lee Oei

Academic Editor

PLOS ONE

Reviewers' comments:

Reviewer's Responses to Questions

**Comments to the Author**

1. If the authors have adequately addressed your comments raised in a previous round of review and you feel that this manuscript is now acceptable for publication, you may indicate that here to bypass the “Comments to the Author” section, enter your conflict of interest statement in the “Confidential to Editor” section, and submit your "Accept" recommendation.

Reviewer #2: All comments have been addressed

2. Is the manuscript technically sound, and do the data support the conclusions?

Reviewer #2: (No Response)

3. Has the statistical analysis been performed appropriately and rigorously? 

Reviewer #2: (No Response)

4. Have the authors made all data underlying the findings in their manuscript fully available?

Reviewer #2: (No Response)

5. Is the manuscript presented in an intelligible fashion and written in standard English?

Reviewer #2: (No Response)

6. Review Comments to the Author

Reviewer #2: Thank you for addressing all my comments. Well done.

Good luck with the further counseling of (future) parents.

7. PLOS authors have the option to publish the peer review history of their article (what does this mean?). If published, this will include your full peer review and any attached files.

Reviewer #2: No

---

## [Author Response · Author response to Decision Letter 1]

2 Jun 2020

Response letter

 Dear academic Editor (Ju Lee Oei)

After going through the entire revised version manuscript, you forwarded your constructive editorial comments which we missed to touch. Therefore, we are glad enough to express our sincerest thanks for your constructive comments that could help improve novelty of our effort.

Editorial comment: The manuscript is much improved. However, please edit it for minor grammatical details and presentation. For example, two decimal points are not needed in many of the results.

Authors’ response: It is really a comment of relevance in increasing readability of the paper through conveying the key messages in an intelligible and easily understandable language. Therefore, the manuscript is again revised for the required grammatical details and presentation of results. Results are now consistently presented with two decimal points which could show higher precision than a single decimal point. 

All the improved changes are incorporated to the second revised version manuscript as shown by the yellow highlighted text in the tracked version of the revised manuscript.

Thank you in advance!

Wubet Alebachew

---

## [Editor Report · Decision Letter 2]

4 Jun 2020

The burden of traditional neonatal uvulectomy among admissions to neonatal intensive care units, North Central Ethiopia, 2019: A triangulated crossectional study

PONE-D-20-01248R2

Dear Dr. Alebachew,

We’re pleased to inform you that your manuscript has been judged scientifically suitable for publication and will be formally accepted for publication once it meets all outstanding technical requirements.

Kind regards,

Ju Lee Oei

Academic Editor

PLOS ONE
---

## [Editor Report · Acceptance letter]

8 Jun 2020

PONE-D-20-01248R2 

The burden of traditional neonatal uvulectomy among admissions to neonatal intensive care units, North Central Ethiopia, 2019: A triangulated crossectional study 

Dear Dr. Alebachew Bayih:

I'm pleased to inform you that your manuscript has been deemed suitable for publication in PLOS ONE. Congratulations! Your manuscript is now with our production department. 

Kind regards, 

on behalf of

Dr. Ju Lee Oei 

Academic Editor

PLOS ONE